# Genome Cytosine Methylation May Affect Growth and Wood Property Traits in Populations of *Populus tomentosa*

**Kaifeng Ma** [1,2,3], **Yuepeng Song** [1,2], **Dong Ci** [1,2], **Daling Zhou** [1,2], **Min Tian** [1,2] **and Deqiang Zhang** [1,2,4,*]

[1] National Engineering Laboratory for Tree Breeding, Beijing Forestry University, Beijing 100083, China; makaifeng@bjfu.edu.cn (K.M.); yuepengsong@bjfu.edu.cn (Y.S.); cdshanx@163.com (D.C.); zag08108042@163.com (D.Z.); minna_tian@163.com (M.T.)

[2] Key Laboratory of Genetics and Breeding in Forest Trees and Ornamental Plants, Ministry of Education, College of Biological Sciences and Technology, Beijing Forestry University, Beijing 100083, China

[3] Beijing Key Laboratory of Ornamental Plants Germplasm Innovation & Molecular Breeding, National Engineering Research Center for Floriculture, Beijing Forestry University, Beijing 100083, China

[4] Xining Forestry Science Research Institute, Xining City, Qinghai 810003, China

[*] Correspondence: DeqiangZhang@bjfu.edu.cn; Tel.: +86-10-6233-6007

**Abstract:** Growth and wood formation are crucial and complex biological processes during tree development. These biological regulatory processes are presumed to be controlled by DNA methylation. However, there is little direct evidence to show that genes taking part in wood regulation are affected by cytosine methylation, resulting in phenotypic variations. Here, we detected epimarkers using a methylation-sensitive amplification polymorphism (MSAP) method and performed epimarker–trait association analysis on the basis of nine growth and wood property traits within populations of 432 genotypes of *Populus tomentosa*. Tree height was positively correlated with relative full-methylation level, and 1101 out of 2393 polymorphic epimarkers were associated with phenotypic traits, explaining 1.1–7.8% of the phenotypic variation. In total, 116 epimarkers were successfully sequenced, and 96 out of these sequences were linked to putative genes. Among them, 13 candidate genes were randomly selected for verification using quantitative real-time PCR (qRT-PCR), and it also showed the expression of nine putative genes of *PtCYP450*, *PtCpn60*, *PtPME*, *PtSCP*, *PtGH*, *PtMYB*, *PtWRKY*, *PtSTP*, and *PtABC* were negatively correlated with DNA methylation level. Therefore, it suggested that changes in DNA methylation might contribute to regulating tree growth and wood property traits.

**Keywords:** growth trait; wood property; cytosine methylation; epimarker; candidate gene; gene expression

## 1. Introduction

In plant species, phenotypic plasticity is a phenomenon that one genotype displays alternative phenotypes induced by changing environments and epigenetic variation [1–4]. Epigenetic regulation is not based on alterations in DNA sequence [5,6]; rather, it gives rise to 'epialleles', meaning alleles with identical DNA sequences yet diverse levels of gene expression resulting from differences in their epigenetic status [7,8]. Research shows that epigenetic changes are potentially reversible, often exist in metastable states [9,10], and can be passed from one generation to the next via mitosis or meiosis

[11–13]. This inherited characteristic provides an opportunity to plot epigenetic linkage maps and perform intraspecific association analysis between epigenetic alleles and traits [14–17].

Cytosine methylation, primarily the addition of a methyl group to the C5 position of a cytosine residue, is one of the best known epigenetic modifications [18]. The modification participates in various important biological responses, e.g., repressing the expression of transposons and repetitive elements, responding to biotic/abiotic stress, and taking part in early embryogenesis, stem cell differentiation, X chromosome inactivation, and genomic imprinting [19–24]. Differences in the extent of DNA methylation can give rise to varied gene expression patterns, leading to phenotypic variation [25]; furthermore, specific sites showing different patterns of 5 mC (either hypermethylation or demethylation) between different individuals or tissues can affect genetic transcription, resulting in morphological changes [26–28].

For some time, researchers focused attention on selected methylated genes and their function in individual plants, such as *Arabidopsis SUPERMAN* genes with cytosine methylation affecting flower development [29], *PHOSPHORIBOSYLANTHRANILATE ISOMERASE* (*PAI*) genes supply insufficient PAI activity with cytosine methylation [30], *DEMETER* genes imprint *MEDEA Polycomb* gene by excising 5-methylcytosine [31], and *FLOWERING WAGENINGEN* (*FWA*) transcription factor gene silenced by DNA methylation in vegetative tissue but it is demethylated in the central cell of the female ovule [13,32]. In recent years, epimarkers were developed to explore epigenetic diversity and structure, and associated genes, on the basis of epialleles that can be detected using the methylation-sensitive amplified polymorphism (MSAP) method in hybrids and wild populations [16,33,34]. Furthermore, these methods led to the notion that germplasm resources carrying epimutations associated with beneficial traits could be selected from plant populations [15–17].

Chinese white poplar (*Populus tomentosa* Carr.), a native tree species with $2n = 2x = 38$ chromosomes (diploid; $n$ and $x$ represent chromosome numbers of gametophyte and haploid, respectively) and widely distributed in the Yellow River basin, plays an important role in ecological and environmental protection. The tree has also been cultivated for commercial timber production and pulpwood because of its low contents of fermentation-inhibiting extractives, as well as high biomass conversion efficiency [35–37]. It has also been used as a model species for perennial woody plants in physiological and biochemical research, genetic diversity and structure analysis, and the investigation of functional genes that participate in lignocellulose biosynthesis and growth processes [38]. Previously, we have explored the genetic/epigenetic diversity and structure of *P. tomentosa* and performed marker–trait association analysis [39,40], as well as showing that candidate wood-related genes with unique methylation patterns play critical roles in xylem biosynthesis [41]. However, epimarker–trait association analysis and seeking functional genes regulated by methylation status have not yet been performed in natural populations of *P. tomentosa*. Therefore, here, we investigate the relationship between global DNA cytosine methylation markers (using MSAP) and variations in tree growth and wood property parameters in order to elucidate the possible epigenetic regulation of xylem formation.

## 2. Materials and Methods

### 2.1. Plant Materials

Plant xylem materials were harvested from the germplasm bank of *P. tomentosa* located at Guanxian County (36°30′54″ N, 115°21′45″ E), Shandong Province, China. The germplasm bank, including 1047 genotypes collected from the region of China in which the species is naturally distributed, was constructed between 1982 and 1984 using root segment propagation techniques [42]. Each genotype was planted with at least three replicates with row spacing 4 m and plant spacing 4 m. In the present study, 432 unrelated genotypes (each genotype has three clones) originating from Beijing (20), Hebei (114), Shandong (19), Henan (114), Shaanxi (64), Shanxi (80), Gansu (6), Anhui (10), and Jiangsu (5) were selected randomly for phenotype measurements and genotyping. To deal with each of the 1296 trees, we first stripped off the bark (approximately 5 cm × 5 cm) at breast height and cut out xylem samples with a sharp blade; each of the xylem samples was divided into two parts

equally, one part of the xylem material was frozen rapidly in liquid nitrogen for nucleic acid extraction and the other part was stored in plastic bags for wood property determination at normal temperature status, respectively.

## 2.2. Phenotypic Data Collection

Phenotypic data, including growth traits and wood properties, were collected and then used in statistical analysis. The growth phenotype traits of tree height (H) and diameter at breast height (DBH) were measured by using the enclosed ruler method and hypsometer, respectively. The volume of timber (V) was estimated according to the following equation [43]: $V = \pi \times (DBH/2)^2 \cdot H \cdot f$ with $f = 0.488$.

The xylem materials that were stored in plastic bags were taken back to the laboratory and dried naturally for determining wood properties, including fiber length, fiber width, microfibril angle (MFA), and contents of lignin, holocellulose, and $\alpha$-cellulose. Firstly, the MFA was measured using an X-ray diffractometer (PanalyticalX'Pert Pro, Philips, Eindhoven, Netherlands) with the angle between the incident ray and the receiving optical path set to 22.4°. The main parameter settings were: tube voltage 40 kV; tube current 40 mA; scanning step 0.5°; and angle of rotation 0–360°. The MFA was calculated using the method of 0.6 T [44,45]. Secondly, we cut a small amount of each xylem sample into small pieces and then macerated the tissue in a solution of hydrogen peroxide (30%) and glacial acetic acid (1:1, *v/v*) at 70 °C for 48 h. For fiber length and fiber width determination, the fibrous material was washed with deionized water, processed with safranin staining, and placed uniformly over a slide, then measured using a computer image analysis system with a color television video camera (VM-60N, Olympus, Japan) according to Hart et al. [46]. Then, the xylem samples were ground into fine powder and passed through a 40–60 mesh screen; later, the contents of lignin, holocellulose, and $\alpha$-cellulose were determined and measured using wet chemistry analysis techniques described by Porth et al. [47]. At the same time, near-infrared reflectance (NIR) spectroscopy was used to detect the NIR absorption spectra of the powder samples, and calibration models were developed to predict the contents of holocellulose, $\alpha$-cellulose, and lignin [48].

The plain vanilla ANOVA (analysis of variance) was used to see the significant difference ($p < 0.05$) in the phenotypic variables among the nine populations. Additionally, we used Duncan's multiple range test to detect the significance difference ($p < 0.05$) within the pairs of means of each phenotypic parameter for the populations [49].

## 2.3. Genotyping by Methylation-Sensitive Amplified Polymorphism

The frozen xylem materials were ground into fine powder in porcelain mortars with liquid nitrogen for the extraction of nucleic acid DNA and RNA. Half of each powder sample was used for DNA isolation by the Cetyltrimethyl Ammonium Bromide (CTAB) method [50], and DNA was quantified using a NanoVue UV/visible spectrophotometer (GE Healthcare Company). We processed the DNA for methylation-sensitive amplified polymorphism (MSAP) detection, including double digestion, ligation, pre- and selective amplification, and capillary electrophoresis with fluorescence detection, as described by Ma et al. [40]. Each DNA sample was digested separately by restricted endonuclease combinations of *Eco*RI–*Hpa*II and *Eco*RI–*Msp*I, and the electrophoresis results for the digested mixtures were analyzed using GeneMarker V1.7.1: four patterns of genotyping were generated (denoted '1,1'; '1,0'; '0,1'; and '0,0') according to the presence ('1') or absence ('0') of the relevant fragment in the electrophoresis lanes of the final amplification products pre-digested by *Eco*RI–*Hpa*II and *Eco*RI–*Msp*I, respectively.

## 2.4. Linear Correlation Analysis and Association Analysis

Four patterns of genotyping were generated from the procedure for MSAP detection, and the percentage of each pattern relative to the total number of bands was defined as relative hemi-methylation ('1,0'), full-methylation ('0,1'), non-methylation ('1,1'), or uninformative site ('0,0'). The total relative methylation level was defined as the sum of hemi-methylation and full-methylation percentages. Pearson correlation analysis was carried out between relative methylation levels and

traits (tree height, diameter at breast height, volume of timber, fiber length, fiber width, MFA, and contents of lignin, holocellulose, and $\alpha$-cellulose) to investigate the relationship between methylation and phenotype. Meanwhile, the relationship between gene expression level and DNA methylation level was also estimated by using linear correlation analysis. Primary percentage data were transformed as $x_{ij}' = \arcsin\sqrt{x_{ij}}$ before calculation ($x_{ij}$ is the *j* observed value in the *i* group).

In order to identify correlations between phenotypic traits and the MSAP markers with four patterns at an epigenetic locus, single marker association analysis was carried out using single-factor ANOVA with a significance threshold of $p = 0.05$ or $p = 0.01$ in general. The contribution to trait variation of each MSAP marker was evaluated as the percentage of the square deviation among groups and population variance [39]. The structural network of associated epimarkers and the traits was constructed using a Cytoscape V3.5.1 software (https://cytoscape.org/what_is_cytoscape.html) according to the instructions.

## 2.5. Candidate Gene Screening and Gene Expression

The associated MSAP markers were separated from the selective amplification products using electrophoresis on 6% denaturing polyacrylamide gels and detected with silver staining [51]. Then, the candidate marker fragments were extracted from the gels with the Wizard SV and PCR Clean-Up System (Promega, Madison, WI, USA), and the short PCR fragments were sequenced by Biomed Company (Beijing, China) after transformation and cloning processes [38]. Sequence homology analysis and function prediction were performed using web databases, including National Center of Biotechnology Information (NCBI) and Joint Genome Institute (JGI).

We performed total RNA isolation using the RNeasy plant mini kit (Qiagen, Shanghai, China) from powdered xylem (mentioned in Section 2.3), and used the first-strand cDNA (synthesized using Reverse Transcription System, Promega) for quantitative real-time PCR (qRT-PCR) processes following the description by Song et al. and Ma et al. [52,53]. Primers were designed according to the functional annotation of the linked genes obtained from the homology sequence alignment. The specificity of each primer set was checked by sequencing the PCR products, and 13 candidate genes were selected for verification using *PtACTIN* as the internal control gene [54]. Primers for qRT-PCR were designed using Primer Express 3.0 software (Applied Biosystems) and were listed in Table S1. All reactions were performed in triplicate as technical and biological repetitions in 45 genotypes (135 trees or clones).

## 3. Results

### 3.1. Variation in Growth and Wood Characteristics

The tree phenotype, including growth traits (diameter at breast height, height of tree and volume of timber) and wood characteristics (fiber length, fiber width, microfibril angle and contents of lignin, holocellulose, and $\alpha$-cellulose) were quantified in natural populations planted in the germplasm bank in Guanxian County. In total, the phenotypic variation was analyzed in 432 genotypes (1296 clones) collected from nine different geographic provenances: it was considered that this range of genotypes should provide substantial materials for selective breeding and genetic improvement. The contents of lignin, holocellulose, and $\alpha$-cellulose were 20.87 ± 2.67% (mean ± SD; SD, standard deviation), 72.59 ± 10.59%, and 40.08 ± 8.78%, respectively. The values of fiber length and width were distributed within the range 0.866–1.512 mm and 16.984–29.850 μm with mean values 1.169 ± 0.085 mm and 23.161 ± 1.973 μm, respectively. The mean microfibril angle was 17.815° ± 4.526°. The commercially important parameters of diameter at breast height (21.38 ± 5.67 cm), the height of the tree (14.57 ± 2.88 m), and the volume of timber (0.29 ± 0.19 m$^3$) were also measured.

We performed one-way ANOVA to investigate the genetic variation in the growth and wood property traits among the nine natural populations (Figure 1). Three parameters of wood properties (contents of holocellulose and $\alpha$-cellulose, and fiber width) showed significant differences among the

populations; similarly, the growth traits of diameter at breast height, the height of the tree, and the volume of timber displayed statistically significant significances ($p < 0.05$).

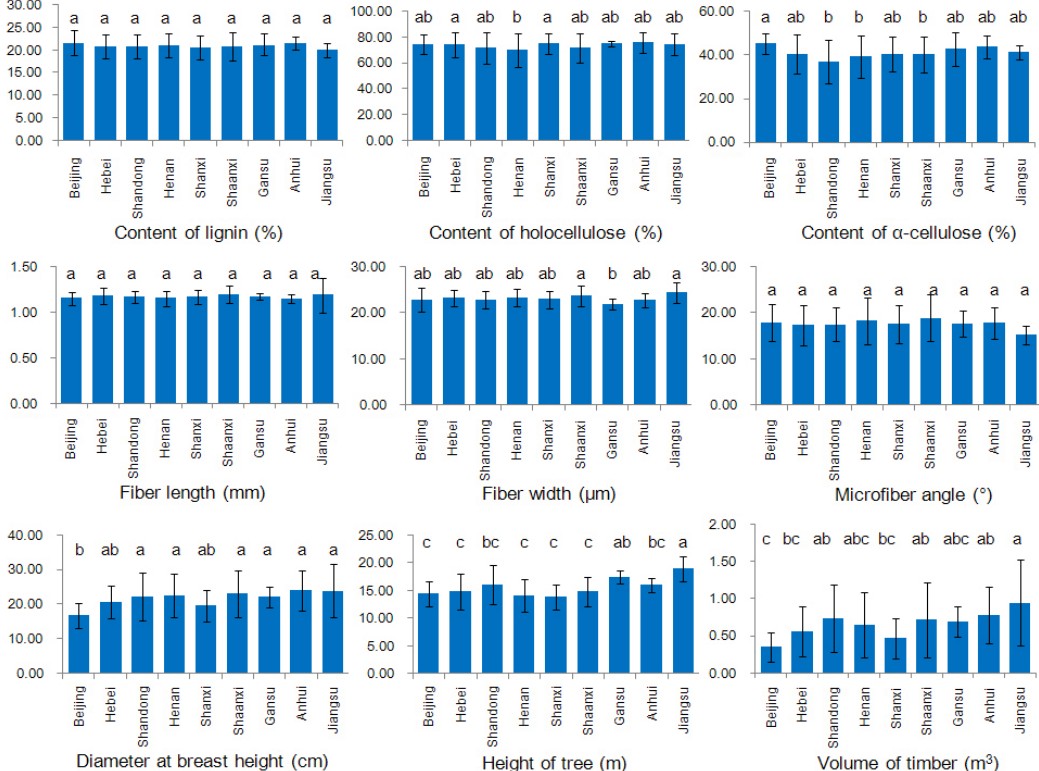

**Figure 1.** Differences in phenotypic traits among nine natural populations of *Populus tomentosa*. The *x*- and *y*-axes indicate provenance and phenotypic value, respectively. Mean ± SD (standard deviation); different letters above the bars (SD) indicate significant differences ($p < 0.05$) according to Duncan's multiple range test (within each sub-graph, it shows a significant difference between each two means if the pair of data sets have no common letter above the bars).

### 3.2. Linear Correlation between PhenoType and DNA Methylation Levels

Linear correlation analysis was performed to determine trait–trait phenotypic correlations and trait—methylation level correlations among *P. tomentosa* genotypes (Table 1). Lignin content was negatively correlated with cellulose (holocellulose, $\alpha$-cellulose) content. Three wood and growth parameters (fiber length, timber volume, diameter at breast height) were significantly correlated with the other traits. For instance, fiber length was positively correlated with fiber width, diameter at breast height, tree height and timber volume; in contrast, fiber length was negatively correlated with microfibril angle ($p < 0.01$). Timber volume was positively correlated with lignin content, diameter at breast height, as well as tree height, but was negatively correlated with $\alpha$-cellulose content ($p < 0.01$). These results suggest that it was possible to select individuals for long fibers and large quantities of stem volume according to measurements of diameter at breast height and tree height.

**Table 1.** Correlations among phenotype parameters and between phenotype and genome methylation/non-methylation levels in the xylem tissue of *Populus tomentosa* [a,b].

| | L (%) | HC (%) | α- (%) | FL (mm) | FW (μm) | MFA (°) | DBH (cm) | H (m) | V (m³) | Non- (%) | CNG (%) | CG (%) | Uninformative (%) |
|---|---|---|---|---|---|---|---|---|---|---|---|---|---|
| HC (%) | −0.229 ** | | | | | | | | | | | | |
| α- (%) | −0.248 ** | 0.611 ** | | | | | | | | | | | |
| FL (mm) | −0.000 | −0.012 | −0.034 | | | | | | | | | | |
| FW (μm) | −0.004 | −0.017 | −0.041 | 0.206 ** | | | | | | | | | |
| MFA (°) | −0.008 | 0.043 | 0.037 | −0.383 ** | −0.093 | | | | | | | | |
| DBH (cm) | 0.159 ** | −0.120 * | −0.181 ** | 0.280 ** | 0.157 ** | −0.114 * | | | | | | | |
| H (m) | 0.083 | −0.001 | −0.095 | 0.246 ** | −0.001 | −0.143 ** | 0.634 ** | | | | | | |
| V (m³) | 0.134 ** | −0.080 | −0.161 ** | 0.273 ** | 0.095 | −0.085 | 0.945 ** | 0.728 ** | | | | | |
| Non- (%) | 0.010 | −0.011 | −0.049 | −0.039 | 0.037 | 0.015 | 0.010 | −0.109 * | −0.049 | | | | |
| CNG (%) | 0.035 | 0.028 | 0.032 | 0.016 | −0.002 | 0.041 | −0.064 | 0.002 | −0.021 | −0.741 ** | | | |
| CG (%) | −0.039 | 0.011 | 0.017 | 0.056 | 0.071 | −0.005 | 0.077 | 0.122 * | 0.096 | −0.413 ** | 0.356 ** | | |

| | | | | | | | | | | | | | |
|---|---|---|---|---|---|---|---|---|---|---|---|---|---|
| CG/CNG (%) | −0.032 | −0.017 | 0.032 | 0.014 | −0.088 | −0.060 | 0.009 | 0.097 | 0.044 | −0.518 ** | −0.103 * | −0.254 ** | |
| Total (%) | 0.013 | 0.026 | 0.032 | 0.035 | 0.027 | 0.030 | −0.019 | 0.051 | 0.022 | −0.748 ** | 0.931 ** | 0.671 ** | −0.181 ** |

[a] Abbreviations: L, lignin; HC, holocellulose; $\alpha$-, $\alpha$-cellulose; FL, fiber length; FW, fiber width; MFA, microfibril angle; DBH, diameter at breast height; H, height of tree; V, volume of timber; Non-, relative non-methylation level; CNG, relative hemi-methylation level; CG, relative full-methylation level; Uninformative, relative level of uninformative site; [b] The primary percentage data were transformed as $x_{ij}' = \arcsin\sqrt{x_{ij}}$ before calculation; * $p < 0.05$ (2-tailed), ** $p < 0.01$ (2-tailed).

We also analyzed the relationships between phenotypic traits and relative gene methylation levels estimated from 2393 polymorphic MSAP markers (epimarkers) out of 2408 bands according to our previous data files [40]. The relative total methylation and non-methylation levels were 26.55% and 42.71%, respectively; the relative hemi-methylation level (13.47%) was larger than that of full-methylation (13.10%) ($p < 0.001$) [40]. Here, it showed that tree height was significantly negatively correlated with relative non-methylation level but positively correlated with relative full-methylation level ($p < 0.05$) (Table 1).

### 3.3. MSAP Markers Associated with Phenotypic Traits within the Populations

Though we investigated the linear correlations between phenotypic traits and relative gene methylation levels, the relationship was unclear between traits and each of the polymorphic epimarker. On the basis of the single marker ANOVA, we considered that 1101 (630 trait-specific and 471 multi-function epimarkers) out of 2393 polymorphic MSAP markers were associated with the nine phenotypic traits ($p < 0.05$). We constructed structural networks (Figure 2) to illustrate their relationships and they showed that 125 (77 trait-specific), 127 (66 trait-specific), and 72 (30 trait-specific) epimarkers, explaining 1.19% ($p = 0.034$) to 4.69% ($p < 0.001$) of variation, were associated with contents of lignin, holocellulose, and $\alpha$-cellulose, respectively (Table S2). Similarly, 1.06% ($p = 0.043$) to 7.15% ($p < 0.001$) of variation in fiber length, fiber width and microfibril angle was explained by 190 (91 trait-specific), 134 (75 trait-specific), and 140 (64 trait-specific) associated epimarkers, respectively. Finally, 312 (281 trait-specific), 414 (254 trait-specific) and 345 (309 trait-specific) epimarkers were associated with diameter at breast height, the height of the tree, and the volume of timber, respectively, explaining 1.13% ($p = 0.035$) to 7.78% ($p < 0.001$) of variation in those traits (Figure 2, Table S2). The analysis suggested that these epimarkers might be associated with important functions in the regulation of complex quantitative traits, including wood property and tree growth.

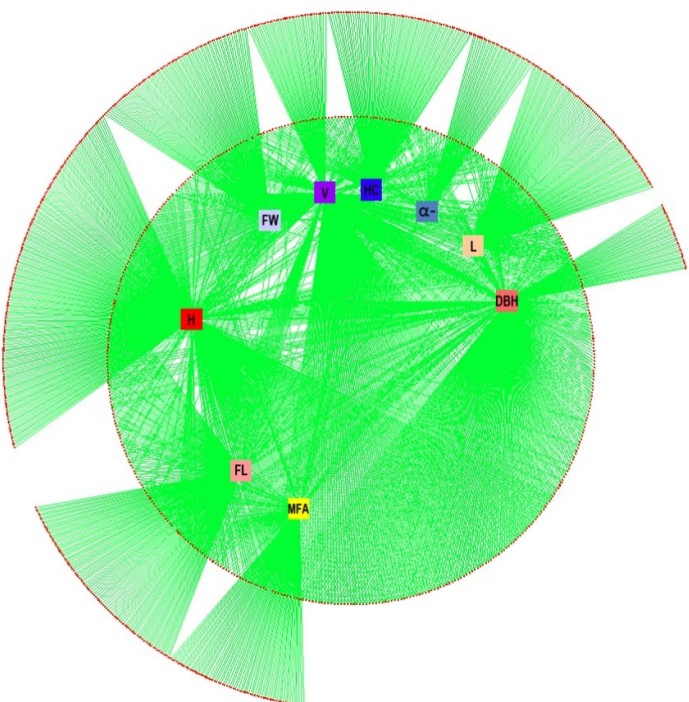

**Figure 2.** Relationships between all of the significant associated epimarkers and phenotypic traits, as represented by a structural network. The square nodes in the innermost circle represent fiber length (FL), fiber width (FW), microfibril angle (MFA), diameter at breast height (DBH), the height of the tree (H), the volume of timber (V), and the contents of lignin (L), holocellulose (HC) and $\alpha$-cellulose ($\alpha$-). Nodes with red around the central circle represent multi-function associated epimarkers. Nodes

with red around the outermost circle represent trait-specific associated epimarkers. The green lines connecting traits with epimarkers represent the phenotype variants explained by the associated epimarkers ($p < 0.05$).

### 3.4. Sequencing and Functional Prediction by Homology Alignment

In order to investigate the function of genes linked with candidate epimarkers that were sequenced by denaturing polyacrylamide gel electrophoresis and silver staining, recovery from the gels, transformation and cloning processes must take place. In total, out of 180 randomly selected MSAP markers, we successfully sequenced 116 epimarker fragments (NCBI, Nos. MN757649–MN757764; Data S1). We aligned the sequences to the reference genome of *P. trichocarpa* and identified 96 markers that were linked to putative functional genes. The linked genes were predicted to take part in, e.g., encoding cytochrome P450 family proteins (MSAP-602), regulatory MYB and WRKY transcription factor families (MSAP-1222, MSAP-2105), ATPase activity regulation (MSAP-803, MSAP-811), regulating transmembrane transport (MSAP-1250), and glycosyl hydrolase family 1 protein (MSAP-2313), as well as other important functions (Table S3). The evidence indicated that the genes linked to epimarkers were essential for plant development, gene regulation via transcription factors, and energy metabolism in *P. tomentosa*.

### 3.5. Quantitative Expression of Candidate Genes

To examine whether the expression of putative genes was influenced by DNA methylation, we investigated the relationship between different patterns of gene exSpression and the relative DNA methylation levels. Thirteen candidate genes were randomly selected for verification using qRT-PCR within 45 genotypes of *P. tomentosa*: We found that the genes of *Potri.002G228400*, *Potri.010G254400*, and *Potri.018G070900* were expressed at high levels (Figure 3a). We then observed that the genes of *Potri.001G223800*, *Potri.002G228400*, *Potri.004G133800*, *Potri.005G091700*, and *Potri.018G070900* were expressed at high levels and showed relatively high non-methylation levels (Figure 3b, in red); however, the genes *Potri.001G223800*, *Potri.005G091700*, and *Potri.018G070900* were expressed at lower levels and showed relatively high hemi-methylation, full-methylation, and total-methylation levels, respectively. Negative correlations were also detected between some of the other genes and DNA methylation levels in different patterns (Figure 3b, Table S4). Homology analysis showed that the nine correlated putative genes were *PtCYTOCHROME P450* (*PtCYP450*, *Potri.018G070900*), *PtCHAPERONES* (*PtCpn60*, *Potri.004G133800*), *PtPECTINMETHYLESTERASE* (*PtPME*, *Potri.015G127500*), *PtSERINECARBOXYPEPTIDASE* (*PtSCP*, *Potri.005G091700*), *PtGLYCOSIDE HYDROLASE* (*PtGH*, *Potri.001G223800*), *PtMYB* (*Potri.015G075800*), *PtWRKY* (*Potri.002G228400*), *PtSUGAR TRANSPORT PROTEIN* (*PtSTP*, *Potri.002G096000*), and *PtATP-BINDING CASSETTE TRANSPORTER* (*PtABC*, *Potri.010G254400*) (Table S3).

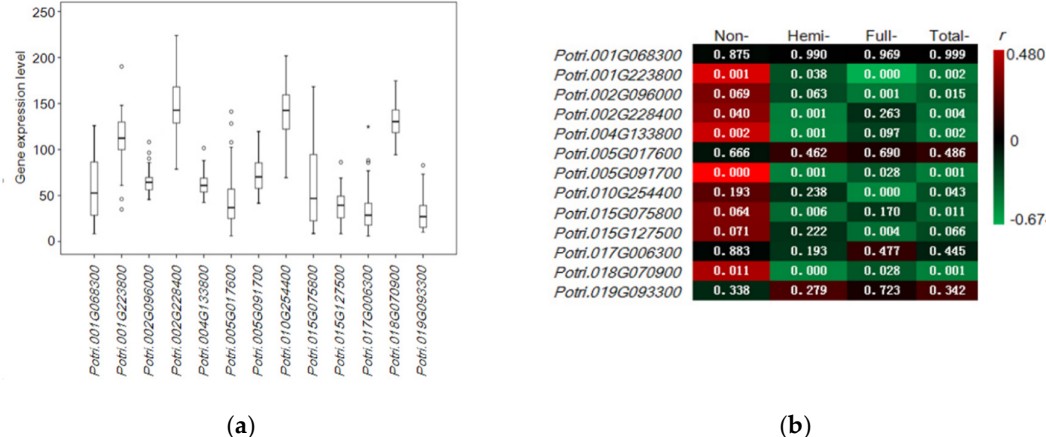

**Figure 3.** Expression levels of candidate genes and their correlations with relative DNA methylation levels. (**a**) Gene expression level determined by quantitative real-time polymerase chain reaction (qRT-PCR). The *x*- and *y*-axes indicate, respectively, gene name and expression level relative to *PtACTIN*. (**b**) Linear correlations between gene expression levels and relative DNA methylation levels. Non-, Hemi-, Full-, and Total- indicate relative non-methylation, hemi-methylation, full-methylation, and total-methylation levels, respectively, estimated by the Pearson correlation coefficient (*r*, red-black-green color scale from positive to negative correlation). The white number (*p*-value) in each cell indicates the significance of difference.

On the basis of previous research, the *Arabidopsis* CYP450 Reductase 2 (ATR2), providing electrons from NADPH to a large number of CYP450 [55], appears to be induced during lignin biosynthesis and under stress [56]. Cpn60 are large double-ring assemblies that assist in the folding of the key proteins in plant growth and development [57–59]. The roles of PME have not been fully elucidated; however, this enzyme may be involved in cell elongation by modifying cell wall pectin [60]. It has long been proposed that *PMEs* have roles in growth regulation, including stem growth [61,62], and *PttPME1* was demonstrated involving in mechanisms determining fiber width and length in the wood cells of aspen trees [63]. The serine carboxypeptidase (SCP) family, also called SCP-like (SCPL) family, plays a key part in plant growth, development and stress responses. Transgenic plants of *Nicotiana tabacum* over-expressing *NtSCP1* show reduced cell elongation [64]. In *Triticum aestivum*, SCP regulates cell death during vascular tissue development [65]. Glycosyl hydrolase (GH) proteins are broadly distributed in organisms, and β-glucosidases belonging to GH family 1 have been implicated in several fundamental processes, including lignification [66,67].

MYB and WRKY are two transcription factor families involved in most biological processes. It was shown that the MYB61, regulated by NAC29/31, binds with *CELLULOSE SYNTHASE*, which in turn activates gene expression in secondary wall cellulose synthesis in *Oryzasativa* [68]. Similarly, *PtrMYB152* is over-expressed in secondary wall-forming cells, resulting in the specific activation of lignin biosynthetic genes in *P. trichocarpa* [69]. It was also reported that PtrWRKY19 and VvWRKY2 may function as regulators of pith secondary wall formation in poplar and lignification in grapevine, respectively [70,71].

Sugar transport protein (STPs) and ATP-binding cassette transporter (ABC) transporters are important for transmembrane transport, an essential biological process in cells. STPs are high-affinity hexose transporters with sink-specific tissue expression [72–74]; constitutive over-expression of *STP13* resulted in seedlings with increased biomass when grown on media supplemented with sugar [75]. It was suggested that *MeSTPs* may play a role in early tuber growth, the period when these genes were mainly expressed in *Manihot esculenta* [76]. A breakthrough study demonstrated that a transporter of the ABC family pumps the monolignol *p*-coumaryl alcohol, one of the three main monolignols synthesized in the cytosol, across the plasma membrane [77,78]. It was also reported that the stem vascular morphology was slightly disorganized in *abcb14-1* mutants, with decreased

phloem area in the vascular bundle and decreased xylem vessel lumen diameter [79]. Therefore, it seemed that cytosine methyaltion might take part in regulating the expression of the putative genes and affect the growth and wood property traits.

## 4. Discussion

Growth traits and wood properties, determining the business value and potential for energy production, are the most important phenotypes for commercial timber. *P. tomentosa* is a native species mainly used for timber in vast regions of North China; thus, it is imperative to investigate the mechanisms of growth regulation and wood formation to underpin the selection of germplasm resources and breeding for genetic improvement. In this paper, we measured and calculated values of nine parameters of phenotypic traits (diameter at breast height, height of tree, and volume of timber; content of lignin, content of holocellulose, content of α-cellulose, fiber length, fiber width, and microfibril angle) among 1296 plants (432 genotypes) from natural tree populations. Then, we explored the relationships between those traits and variation in DNA methylation detected using MSAP and qRT-PCR methods.

On the basis of the statistics of the different genotypes, we found that the coefficient of variation in traits was 7.24–66.91%, and the growth and wood property traits showed significant variation among the nine natural populations, with the exception of lignin content, fiber length and microfibril angle. This phenotypic variation, or plasticity, might correlate with genome DNA methylation, the level of which plays an important role in genome stabilization and transposable element repression [19,22]. Previous evidence showed that the methylation level is correlated with phenotype [80]. For instance, a negative correlation was detected between genomic methylation level and gene expression in maize; similarly, negative correlations were reported between methylation level and energy use efficiency and crop yield in *Brassica napus* [80,81]. However, in the woody ornamental plant mei, the leaf length, width, and area were positively correlated with relative full and total methylation levels [17]. In hybrids of poplar, a positive correlation was demonstrated between DNA methylation percentage and productivity [82]. We previously showed that the net photosynthetic rate, tree height and diameter at breast height were positively correlated with relative total methylation and hemi-methylation levels in an intraspecific hybrid population of *P. tomentosa* [39]; here, we found a similar result among the natural populations—the relative full-methylation level was positively correlated with tree height.

Previous studies have showed that methylation is essential for gene exploration and epimarker-assisted analysis in plant populations [15,17,34]. For instance, *PtHT1.1*, *PtHT1.2*, *PtPsbK*, *PtPIN1.2*, *PtMYB60*, and *PtMYB61* were found to be modified by DNA methylation and regarded as playing roles in leaf formation and regulation of photosynthesis in *P. simonii* [34]. In our research, we detected 1101 epimarkers associated with growth and wood property traits. From these, we identified 96 epimarker-linked putative genes, 13 of which were randomly selected for qRT-PCR analysis and it was found that *PtCYP450*, *PtCpn60*, *PtPME*, *PtSCP*, *PtGH*, *PtMYB*, *PtWRKY*, *PtSTP*, and *PtABC* were negatively correlated with relative DNA methylation level. Meanwhile, it was demonstrated that *PttPME1* involved in mechanisms determining fiber width and length in the wood cells of aspen trees [63]. The transcription factor gene *PtrMYB152* is over-expressed in secondary wall-forming cells, resulting in the specific activation of lignin biosynthetic genes, and PtrWRKY19 may function as regulators of pith secondary wall formation in *P. trichocarpa* [69,71]. The *ABCB14-1* plays a role in decreasing the phloem area in the vascular bundle and xylem vessel lumen diameter in *Arabidopsis* [79]. The above evidence seems to suggest that genes involved in regulating growth and wood development in *P. tomentosa* might be affected by cytosine methylation modifications. The epimarkers that needed to be verified may also provide a new sight for breeding programs in this commercially important tree species.

## 5. Conclusions

The quantitative traits of the growth and wood property are important for commercial timber tree species. The objective of the present study was to reveal the relationship between tree growth, wood property traits and cytosine methylation, respectively. It was found that the tree height was positively correlated with relative full methylation level. And 1101 single- and multi-function MSAP markers explaining 1.1–7.8% of phenotypic variation were associated with growth traits (diameter at breast height, the height of the tree and the volume of timber) and wood characteristics (fiber length, fiber width, microfibril angle, lignin, holocellulose, and $\alpha$-cellulose). It was demonstrated that 96 sequences, out of 116 successfully sequenced epimarkers, were linked to putative genes. The expression levels of nine putative genes (*PtCYP450*, *PtCpn60*, *PtPME*, *PtSCP*, *PtGH*, *PtMYB*, *PtWRKY*, *PtSTP*, and *PtABC*) were negatively correlated with DNA methylation. Our results imply that the widespread natural variation of DNA methylation might contribute to regulating tree growth and wood formation, and the findings will enhance our understanding of epigenetics in tree growth and xylem formation.

**Supplementary Materials:** The following are available online at www.mdpi.com/1999-4907/11/8/828/s1, Table S1: Primer sequences for Real-time PCR, Table S2: MSAP markers associated with phenotypic traits and the variation explanation in natural population of *Populus tomentosa*, Table S3: Homologous alignment and gene function analysis, Table S4: The gene expression levels detected using qRT-PCR and relative cytosine methylation levels within 45 genotypes of *Populus tomentosa*. Data S1: Sequences for the 116 MSAP markers.

**Author Contributions:** D.Z. (Deqiang Zhang) designed the experiments. K.M., Y.S., and D.C. collected the plant materials and phenotypic data. K.M., D.Z. (Daling Zhou), and M.T. did the experiment in gene cloning and quantification of gene expression. K.M. detected the molecular markers and analyzed all of the data profiles. K.M. and D.Z. (Deqiang Zhang) wrote the manuscript. Y.S., D.C., D.Z. (Daling Zhou), and M.T. provided suggestions for manuscript revision. All authors have read and agreed to the published version of the manuscript.

**Funding:** This work was supported by the National Natural Science Foundation of China (Nos. 31872671 and 31670333), and the 111 Project (No. B20050).

**Conflicts of Interest:** The authors declare no conflict of interest.

**Sequencing Data:** All of the sequences were submitted to NCBI (https://www.ncbi.nlm.nih.gov/) with accession numbers: MN757649–MN757764.

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
