# Peer review of "Genome Cytosine Methylation May Affect Growth and Wood Property Traits in Populations of Populus tomentosa"

_forests, doi:10.3390/f11080828_

Round 1

Reviewer 1 Report

Comments and Suggestions for Authors

In the manuscript entitled " : Genomic methylation is associated with variation in growth and wood property traits in populations of Populus tomentosa " the authors studied how genomic methylation affects growth on Populus tomentosa trees using a methylation-sensitive amplification polymorphism (MSAP) method and qRT-PCR to quantify nine putative genes with possible important biological functions.

Minor revision

- The title should be more representative of the text, shorter and easier to read.

- The English needs to be revised. For ex. In the introduction, the first sentence ‘’ Phenotypic plasticity in plant species, induced by epigenetic variation and changing environments [1-3], is epigenetic in that one genotype displays alternative phenotypes under different conditions [2,4]. ‘’should be simplified. It is not clear.

- Defined and discuss briefly the cytosine methylation related to the function of selected the methylated genes in the introduction (PAI, FWA, …)

- The introduction should be revised to better reflect the content and results proposed in this article.

- "we froze part of the xylem materials rapidly in liquid nitrogen for nucleic acid extraction and stored the other part of the xylem materials in plastic bags for wood property determination at normal temperature status". What do you mean normal temperature status? Precise.

- Figure 4 legend add a space to chainreaction

- Figure 4 please define how many cytosine are methylated when referring to hemi- full and total, it is not clear.

- In the qRT-PCR experiment, why did you use a single internal control gene (PtACTIN)?

- In the discussion, "Previous evidence showed that the methylation level is correlated with phenotype. " please add the reference.

- "For instance, Long et al. [14] constructed a new linkage epi-map with both epimarkers and markers in Brassica napus, and identified 125 additional significant quantitative trait loci associated with seven important agronomic traits". what is the link between this sentence and the results of this search?

- The author discusses the CYP450, Cpn60, MYB and SCP only in the results and the discussion. Why not talk about them previously on the introduction before the results and discussion?

- From these we identified 96 epimarker-linked genes with known functions, 13 of which were randomly selected for qRT-PCR analysis. We found that putative genes PtCYP450, PtCpn60, PtPME, PtSCP, PtGH, PtMYB, PtWRKY, PtSTP, and PtABC, the functions of which are discussed below, were negatively regulated by DNA methylation.

The function of these gene is not known specifically related to the tree traits study. Please discuss and related the genes selected with the trait.

- In the atr2 mutation, the total lignin amount in the main inflorescence stem was reduced by 6% and the cellulose was more susceptible to enzymatic hydrolysis after alkaline pretreatment [56,59]. More markedly than disrupting ATR2, disrupting Cytochromeb5 Isoform D (CB5D) controlling syringyl (S)-lignin biosynthesis resulted in >60% reduction in Slignin subunit levels [60]. It is not clear what is the relevance of this with the results obtained here. Is the sequence of PtCYP450 negatively regulated here has any correlation or link with ATR2-like genes in trees? Does this regulation exist in populous ?

- "The SCP family, also called SCP-like (SCPL) family, plays a key part in plant growth,

development and stress responses. Transgenic plants of Nicotiana tabacum over-expressing

NtSCP1 show reduced cell elongation [68]. " This should be moved to the "Introduction" or "Materials and Methods".

- Note: If the article is about genomic methylation, why not talk about acetylation and histones?

- The evidence does not strongly suggests that genes with cytosine methylation modifications are involved in regulating growth and wood development in P. tomentosa. All is speculation in my opinion.

How is the finding related to timber traits? How can it be related to forest industry ? How is it related to business value and potential for energy production ?

Author Response

Dear reviewer, 

Best wishes.

Ma Kaifeng

Reviewer 2 Report

This manuscript analysis a specific plant species Populus tormentosa and relation of its phenotypic traits with epimarkers. Potentially it could be very useful and interesting study. However the explanation of the analysis is fragmented and disconnected and in the end it leaves the reader wandering what were the actual value and results of this study.   There are several big issues that this study has. First of all I was not able to find and verify the sequences with the provided accessions MN757649–MN757764 that were indicated as submitted to the NCBI. And the second big issue - there are many methodological mistakes, omissions, discrepancies and barely supported claims. At this stage and in this respect the manuscript needs rigorous description of methods without mistakes, so that the presented results are supported by a strong evidence. Without such , the numbers and analysis presented in this study are questionable.   Other comments:
Page 2  - "(Populus tomentosa Carr., 2n = 2x = 38)," - please define meaning of n, x and  2n=2x=38)
  Page 4 Paragraph 3.1 - "the stem volume demonstrated great genetic variability (cv = 66.91%) " - please define and explain a measure cv - how exactly this measure demonstrates a genetic variability or provide a reference.   Page 2 Paragraph 2.1 and Page 4 Paragraph 3.1 - " 432 unrelated genotypes with three clones originating" and " In total, the phenotypic variation was analyzed in 432 genotypes (1,296 clones) collected from nine different geographic provenances" - the statement of 432 genotypes is confusing. Please clarify what is meant by  432 unrelated genotypes ? What makes a genotype in these 432 samples/examples? How the genotype is defined in this data?    In Figure 1 - some diagrams do not really resemble a Normal distribution. In addition to the frequency, please provide a plot of theoretical probability distribution density with the estimated parameters of Mean and SD.   Page 5 - "We performed one-way ANOVA to investigate the genetic variation in the growth and wood property traits among the nine natural populations" - Please provide a table showing how many samples were analyzed in each population: Beijing, Hebei, Shadong, Henan, Shanxi, Shaanxi, Gansu, Anhui, Jiangsu.   Page 5 - "Duncan’s multiple range test" - Please provide a reference to the method.   In Figure 2 - "different letters above the bars indicate significant differences according to Duncan’s multiple range test, P < 0.05" - meaning of the letters should be explained. Otherwise it is not clear what Figure 2 shows.   Related to the Figure 2 "The populations from Beijing and Jiangsu showed the lowest and highest mean values for the growth traits, respectively, according to the Duncan’s multiple range test."  - Looking at the Figure, only a few traits seem to be different across provinces. Please explain what do you exactly mean by LOWEST and HIGHEST values and why it is important to mention that.    Page 6 "the relative hemi-methylation level (13.47%) was larger than that of full-methylation (13.10%) (P < 0.001) [39]." - What is the purpose for this reference here? Looking at the numbers I feel suspicious how the 13.47% might be different from 13.10% at the level of significance P < 0.001 ?  To prove that this is really a case you have to provide full details of the method of the statistical test,   and sample sizes ,  and analysis of the statistical power (you can use GPower) .  The same place " This calculation seemed that tree height showed a similar tendency to gene full-methylation level (Table 1)" - confusing sentence, not clear what is meant here.   Page 7 "We constructed structural networks (Figure 3) to illustrate their relationships and they showed" - first of all the construction of the structural networks is not explained in the Methods while it should be explained there. Therefore for the rest of the paragraph 3.3 it is not clear what results are discussed. Specifically, since the construction of the networks is not explained, it is not clear how the structural networks show the  associations between epimarkers and traits. In the same place- "125 (77 trait-specific), 127 (66 trait-specific), and 72 (30 trait-specific) epimarkers, explaining 1.19% (P = 0.034) to 4.69% (P < 0.001) of variation" - what these P-values stand for? What is method and metrics is used here to explain variation? If these values come from ANOVA, it should be clear in the text.   Figure 3 - It does not add anything to this manuscript and doesn't show any particular detail. Even If I zoom very closely, I can't discern what is written in the nodes with red color. It is obsolete here. If this association information is important, then I would suggest to put it into the correspondence tables as a supplementary table.   Page 8. "We aligned the sequences to the reference genome of P.trichocarpa" - Please explain the rationale for choosing this reference genome since it is different from the species you are working with.   Page 8 "NCBI, Nos. MN757649–MN757764" - I have searched NCBI for these accessions, there were 0 results. Please explain and provide correct accessions. The accessions referred to and absent from the database  invalidate the rest of the content in this paragraph.
Page 8 Paragraph 3.5 - What is intended to show with this paragraph? Since it is based on the thirteen randomly selected candidate genes (out of 96 putative genes)  - their in depth associations and analysis has very little of explanatory value.  The correlations in Figure 4 do not seem very different between expression of the particular gene and methylation levels except of few. More- the color scale seems to be disconnected from the actual correlation table - note the bright red cell with the number 0.000 and color of 0 in the color scale bar next to the figure.  Such discrepancies put in question all analysis that is presented in this paper.    

Author Response

Dear reviewer, 

Best wishes.

Ma Kaifeng

Round 2

Reviewer 2 Report

Thank you for answering my comments. However, some of the comments are nor answered satisfactorily. The reference [49] is not a reference explaining the Duncan's statistical method, but a reference to he work in which the Duncan's method was used. In this case, still you have to explain the essence and rationale of the Duncan's method , since there are other statistical multiple comparisons tests.

On the same note, still, the meaning of lowercase letters in Figure 1 is not explained , what exactly means a, b, c,  ac , what these letters show. If they can't be explained, then why they are presented at all? If it is significance it should be clear, what the individual letters mean and what the combinations mean. Such omissions diminish a value of the presentation. 

Similarly the Figure 3 ( 43 in the revised version)  was not satisfactorily addressed. The b) label is missing. The color scale shows correlations from negative to positive this is clear, but the white numbers in the cells remain confusing and suspicious. What does it mean a 0.000 or 0.999 significance of the difference( from the Figure caption) ?  If you display a correlation, what difference are you referring to in this figure with these numbers?. Please explain correctly. 

As for sequences submitted to NCBI. Since they are not available at the time of review and publication , to make them available  for the reviewers and the potential readers it is desirable that the sequence data would be shared in a public repository such as zenodo.org or similar. 

Author Response

Dear Reviewer,

Thank you for your comments. Please see the attachment to find the answers to the comments and queries. 

Best wishes.

Ma Kaifeng
